# A systematic review of military-to-civilian transition, The role of gender

**Alexandria Smith**[1]*, **Laura Rafferty**[1], **Bethany Croak**[1,2], **Neil Greenberg**[1], **Rafiyah Khan**[1], **Victoria Langston**[1], **Marie-Louise Sharp**[1], **Anne Stagg**[2], **Nicola Fear**[1,3], **Sharon Stevelink**[1,2]

**1** King's Centre for Military Health Research (KCMHR), King's College London, London, United Kingdom, **2** Department of Psychological Medicine, Institute of Psychiatry, Psychology and Neuroscience, King's College London, London, United Kingdom, **3** Academic Department of Military Mental Health, King's College London, London, United Kingdom

* alexandria.smith@kcl.ac.uk

## Abstract

### Background

The military-to-civilian transition can be a challenging period for many service members; however, recent research suggests that female ex-service personnel (veterans) confront additional complexities during reintegration into civilian life. This systematic review aimed to identify and synthesise findings across qualitative studies exploring the impact of gender on this transition process.

### Methods

Peer-reviewed literature was drawn from a multi-database search, limited to qualitative studies. The studies included either female veterans or both male and female veterans aged 18 years or older who had previously served in the Armed Forces within the Five Eyes (FVEY) countries (Australia, Canada, New Zealand, the United Kingdom, and the United States). We used a Framework Analysis approach to guide the synthesis of the qualitative data. An assessment of study quality was conducted using the Joanna Briggs Institute (JBI) Qualitative Critical Appraisal Checklist for Qualitative Studies. The study protocol is registered with the Open Science Framework (registration: osf.io/5stuj).

### Results

In total, 10,113 articles were screened after the removal of duplicates, 161 underwent full-text review, with 19 meeting the eligibility criteria. The review identified eleven themes split across individual's experience whilst serving and after transitioning out of the military service. Both male and female veterans discussed a period of acculturation when they joined service and adapted to military norms, culture and identity. Female veterans faced additional challenges at this stage centred on the conflict between feminine norms and the military masculine ideal. Upon leaving service both male and female veterans experienced a loss of military identity and purpose, and dissonance with civilian norms illustrating a military-civilian divide. For female veterans, adjustments and adaptations learned in the military

**Data Availability Statement:** All relevant data are within the manuscript and its Supporting Information files.

**Funding:** This project was supported by a grant from the Forces in Mind Trust (FiMT) 2202 The funders had no role in study design, data collection and analysis, decision to publish, or preparation of the manuscript.

**Competing interests:** The authors have declared that no competing interests exist.

clashed with civilian feminine norms and stereotypically male veteran culture. Female veterans also struggled with the legacies of gender inequality, discrimination, and sexual assault which affected their development of a female veteran identity and affected the provision of services designed to meet their needs as a female. Despite these challenges, female veterans' expressed pride in their service and accomplishments.

## Conclusions

Any effort to improve the military-to-civilian transition should take account of the legacy of gender discrimination, especially within the military service, and the potential mismatch between historical civilian female norms and the more traditionally masculine norms of military life.

## Disclosures

This project was supported by a grant from the Forces in Mind Trust (FiMT) 2202.

## Background

Transitioning from the Armed Forces (AF) to civilian life can be challenging for many ex-service personnel (veterans), with many veterans reporting some difficulty integrating into the civilian environment [1]. The loss of their military identity, and the attendant isolation following separation from their military community, can impede service members from fully integrating, or accessing support [2–4]. A negative transition out of the military is associated with a number of undesirable outcomes, including economic and housing instability, family conflict, poor physical and mental health, and reduced well-being [5]. In contrast, the adaptation of one's military identity to civilian cultural values and norms has been shown to improve the transition experience [6–9].

The number of women in the AF is projected to rise over the next decade due to the expansion of roles and targeted recruitment and retention efforts [10–14]. Currently, women account for 11% to 18% of the regular AF within the Five Eyes (FVEY) (Australia, Canada, New Zealand (NZ), the United Kingdom (UK), and the United States (US)) [10,11,13–15]. Despite this increase, most research on the transition process has focused on men [16]. Yet, even with the lag in research among female compared to male veterans, this is quickly becoming a burgeoning research area [17]. Much of the current evidence is drawn from the US and suggests female veterans confront additional complexities in the military-to-civilian transition compared to male veterans and experience worse physical and mental health outcomes [17–19]. The experience from other FVEY countries is less clear [20]. Governments in the FVEY countries recognize the importance of improving the transition experience for male and female veterans and have begun implementing supportive policies [10,21–24]. However, there remains ample room for improvement in policies and programs to support female veterans. We focused our analysis on the Five Eyes alliance, with their shared language, intelligence, and closely aligned military policies, providing a cohesive context for examining veteran transition.

These nations recognize their obligation to support veterans and families, ensuring they are not disadvantaged by their service and that disparities related to service-induced injuries are mitigated. While most of the Five Eyes (FVEY) nations adhere to this principle, the UK has formalized this commitment in the Armed Forces Covenant [25]. As women's representation

in the military grows, understanding whether they face additional challenges during transition is critical to ensure they do not bear undue burdens. Moreover, improving transition support for female veterans can enhance the Armed Forces' retention and recruitment efforts.

To understand and improve the transition experience of female veterans, we need a deeper understanding of how women's military experiences and post-transition experiences may differ from those of men.

Previous scoping reviews of the role of gender in the military-to-civilian transition primarily identified quantitative studies related to physical and mental health outcomes [16,26]. The authors of these scoping reviews noted a rapid rise in the number of publications in the recent decades (1990–2015) [26]. In contrast to this quantitative, or mixed methods focus, the proposed study will conduct a synthesis of the current *qualitative* literature examining the *experience* of military-to-civilian transition, and the impact of gender on this transition. While previous quantitative studies have provided valuable insights into the prevalence of physical and mental health issues and related health services, qualitative studies can offer a deeper understanding of the lived experiences of veterans [26]. This synthesis will contribute to the current literature by illuminating the unique challenges, perspectives, and needs of female veterans during their transition to civilian life.

## Methods

### Information sources and search strategy

A comprehensive search strategy was designed in consultation with a data librarian. The study protocol was registered with the Open Science Framework (registration: osf.io/5stuj) [27]. Full ethical clearance was granted by the Health Faculties Research Ethics Subcommittee, King's College London. Project Reference: HR/DP-22/23-33303. A multi-database search was conducted of Medline, Embase, PsycINFO, Pubmed, Global Health, Web of Science, and EBSCO. The search strategy included controlled vocabulary terms and keyword searches using variations of the following search terms; 'females,' 'veterans,' and 'transition'. Additional detail regarding the search terms and the number of articles retrieved is provided in the supplemental material (S3 File). Reference sections of articles were also cross-checked for further articles of relevance. Our original comprehensive search was conducted in February 2023 and an update search in February 2024 to ensure that the results reflected the most current evidence.

### Inclusion and exclusion criteria

Papers were included if they were peer-reviewed, qualitative or mixed methods studies, and included female participants, age 18 years or older, who had previously served in the AF within the FVEY (Australia, Canada, NZ, the UK, and the US). We included only studies describing the experiences of individuals transitioning from the Armed Forces to civilian life. Studies examining reintegration after deployment, where participants remained in the military, were excluded. We selected FVEY nations for our study due to their comparable military structures and shared intelligence networks. These countries tend to adopt similar policy approaches and maintain all-volunteer forces, allowing for a better comparison The definition of 'veteran status' was determined by the individual authors and these definitions likely varied by across nations [25]. Papers were excluded if they reported solely on male participants or where gender was not identifiable, were not written in English, where not qualitative or mixed methods studies or were inaccessible in full text at the time of the review. There was no restriction placed on the publication date.

## Study selection

Duplicate studies were removed first by EndNote [28], followed by manual checks. The selection of studies occurred in three stages: title; abstract; and full-text review. The initial screening of titles and abstracts was conducted by one reviewer (AS) due to the large volume of studies. To mitigate potential bias, we implemented a conservative approach, predominantly eliminating studies only if they clearly lacked qualitative data or unambiguously met exclusion criteria. This method balanced efficiency with thoroughness, while subsequent stages involved multiple reviewers, ensuring a comprehensive selection process. Two reviewers (AS, LR) independently conducted full-text reviews with consensus required for study inclusion.

An initial search was run in February 2023, and an updated search was conducted in February 2024, which resulted in 10,113 articles after the removal of duplicates. The title and abstract review excluded 9,952, leaving 161 for full text review. Of these, 19 studies met eligibility criteria (Fig 1). Two reviewers (AS and BC) independently conducted an assessment of study

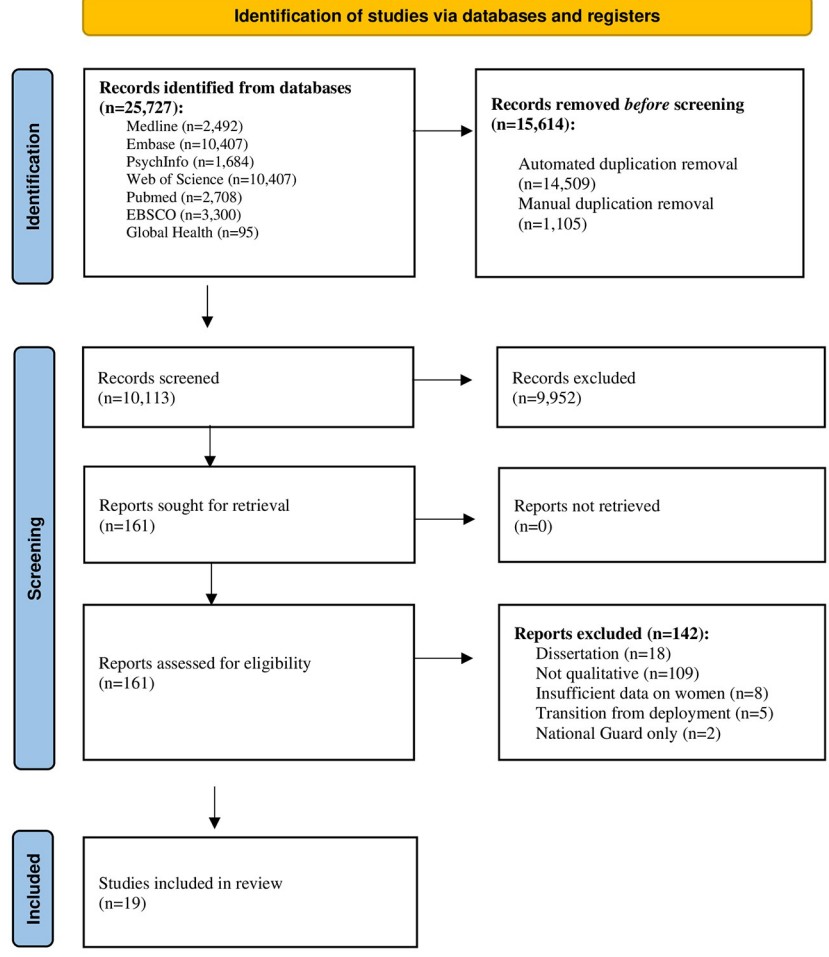

*From:* Page MJ, McKenzie JE, Bossuyt PM, Boutron I, Hoffmann TC, Mulrow CD, et al. The PRISMA 2020 statement: an updated guideline for reporting systematic reviews. BMJ 2021;372:n71. doi: 10.1136/bmj.n71
For more information, visit: http://www.prisma-statement.org/

**Fig 1. PRISMA flow diagram.**

quality using the Joanna Briggs Institute (JBI) Qualitative Critical Appraisal Checklist for Qualitative Studies [29] which is provided in the supplemental material.

## Data extraction

Key study details were extracted into a standardized form, including author(s), year of publication, country, participant characteristics, interview type and analysis. A summary of extracted information is provided in Table 1.

## Synthesis

A Framework Analysis approach was used to guide the synthesis of the qualitative data extracted from the studies [54]. Relevant data was drawn from the original authors' interpretation of the results, the discussion section, and direct citations from participants.

Framework Analysis involves the development of a matrix output where rows and columns are used to summarise cases (in this case qualitative studies) and codes to summarise data. A Framework Analysis typically follows seven stages, the latter six of which were utilized in this research: familiarisation; coding; developing an analytical framework; applying the analytical framework; charting data into the matrix; interpreting the data [54].

As we were extracting narratives from peer-reviewed studies, we bypassed the first stage of Framework Analysis, the transcription of verbal data into text. Each study included in the review was read for familiarisation, with comprehensive notes taken (AS). The *Results* and *Discussion* sections were carefully read line by line with codes (para-phases or labels) assigned to text as appropriate (AS/LR). During this process, an iterative approach was employed with frequent discussions among researchers to ensure consensus in the interpretation of the findings (AS/LR). After coding several papers, LR and AS met and compared codes to create a unified set of codes and to develop a tree diagram to group the codes together. Several iterations of this analytical framework were created until all the studies included in the review had been analysed. Once the framework was finalized, studies were re-read, indexed, and charted onto a matrix displaying the analytical framework. The results of the analysis were explored, and broader concepts around the process of transition were developed.

## Results

### Descriptive results

Nineteen papers were included in the full review with the characteristics of each study detailed in Table 1. No studies were excluded due to concerns with quality (S1 File). Most studies were from the US (n = 14) [2,3,7,33,55–64]; one study included participants from both the US and Israel [65]. The UK contributed two studies [66,67] with Canada and Australia each with an additional study [68,69]. A total of n = 502 veterans were included in the qualitative studies, of which n = 290 were female. The participants ranged from 22 to 71 years of age, with a length of service between 2 to 30 years. The military characteristics of studies varied, including rank (enlisted and officer), branch, and occupational specialties. Most female participants had been deployed at least once. Studies were published from 2013 through 2024, with more than half of the studies (n = 11) published after 2020.

The included studies implemented a variety of analytical frameworks which informed their interpretation of the qualitative data. The most frequent analytic frameworks used included thematic analysis (n = 9) [2,3,55,57,59,63,69–71], grounded theory (n = 5) [33,60–62,68], and phenomenological methodology (n = 6) [7,56,58,64,67,70]. Studies also employed a critical feminist approach (n = 1) [35] or feminist narrative analysis (n = 1) [65]. The studies generated

Table 1. Primary characteristics of studies included in the systematic review.

| Author Publication Year | Country | Participant Characteristics | Primary Underlying Theories | Method of Inquiry | Type of Qualitative Analysis |
|---|---|---|---|---|---|
| Raabe et al. (2024) [63] | USA | Women (n = 3), Men (n = 13) *Characteristics for women only* Age Range: 31–34 Length of service: 8–12 Months since discharge: 3–28 Branch: Army (n = 2), Marines (n = 1), | Self-determination theory [30] | Semi-structured in-depth interviews | Thematic analysis |
| Barrington et al. (2023) [67] | UK | Women (n = 1), Men (n = 4) *Characteristics of full sample (men and women)* Age Range: 35–55 Branch: Army (n = 5) Length of service: 5–20+ years Years since discharge: <1–20 | Social identity theory[31] | Semi-structured in-depth interviews | Phenomenological methodology |
| Murray & Cancio (2023) [64] | USA | Women (n = 1), Men (n = 5) *Characteristics of full sample (men and women)* Age Range: 30–55 years, average = 42.7 Branch: Army (n = 4), Navy (n = 1) Marine (n = 1) Length of service: 5–28 years, average 12.6 years | Community reintegration[32] | Semi-structured in-depth interviews | Phenomenological methodology |
| Rattray et al. (2023) [71] | USA | Women (n = 8) Age Range: 24–53 years, average = 34 Years since discharge (average): 34 | Discovery-oriented approach[33] | Semi-structured interviews at baseline, month 6,12,18 and 24 | Grounded theory, inductive thematic analysis |
| Barnett et al. (2022) [69] | Australia | Women (n = 6) Men (n = 34) *Characteristics of full sample (men and women)* Age Range: 25–57, average = 37 Length of service: 4–40 years, average = 15.1 Branch: Army (n = 25), Navy (n = 7), Air Force (n = 6), Army and Airforce (n = 1), Army and Navy (n = 1) Reason for leaving discharge: Own request (n = 24), Medical discharge (n = 11), Compulsory age retirement (n = 1), Other (n = 4) | Social identity theory [31] and Self-categorization theory [34] | Semi-structured in-depth interviews | Thematic analysis |
| Eichler (2022) [68] | Canada | Women (n = 33) Age Range: 27–64, average = 47 Length of service: 2–39.5 years, average = 19 years Branch: Air Force (n = 10), Army (n = 7), Navy (n = 5), combination (n = 11) | Critical feminist approach with gender as primary category of analysis [35,36] | Semi-structured, in-depth interviews and focus groups | Grounded theory and critical feminist approach |
| Guthrie-Gowerm & Wilson-Menzfeld (2022) [66] | UK | Women (n = 5), Men (n = 6) *Characteristics for women only* Age range 41–72, average 55.4 Length of service: 7–30 years, average 14.4 years Years since discharge: 15–27 years, average 18.8 years | Loneliness through the Social Needs Approach and the Cognitive Discrepancy Model[37] | Semi-structured in-depth interviews | Phenomenological methodology and thematic analysis |

*(Continued)*

**Table 1.** (Continued)

| Author Publication Year | Country | Participant Characteristics | Primary Underlying Theories | Method of Inquiry | Type of Qualitative Analysis |
|---|---|---|---|---|---|
| Laferty et al. (2022) [70] | USA | Women (n = 10)<br>Age Range: 51–85, Average = 59<br>Branch: Army (n = 7), Army National Guard (n = 1), Marine Corps (n = 1), Air Force (n = 1) | None stated | Semi-structured in-depth interviews | Content analysis and inductive thematic analysis |
| Boros et al. (2021) [56] | USA | Women (n = 4)<br>Age Range: 40–60, average = 46.5<br>Branch: Army (n = 2), Air Force (n = 2) | Goffman's Total Institution [38]; Overarching Feminist Theory[39] | Semi-structured, in-depth interviews | Interpretative phenomenological analysis |
| Daphna-Tekoah et al. (2021) [65] | USA / Israel | Women (n = 20), United States military (n = 10), Israel Defence Forces (n = 10)<br>All women completed combat or combat support service | Interdisciplinary: Feminist Security Studies (FSS), International Relations, Guide to Listening[40,41] | Semi-structured, in-depth interviews [†] | Feminist narrative analysis |
| Sayer et al. (2021) [57] | USA | Women (n = 50), Men (n = 50). *Characteristics of full sample (men and women)*<br>Average age = 40.1<br>Military component: Active duty (n = 50), Reserve/Guard (n = 50)<br>Military Branch: Army (n = 58), Air Force (n = 19), Navy (n = 16), Marine Corps (n = 7)<br>Rank: Enlisted (n = 83), Officer (n = 17) | Belonging and social connectedness [42]; Ecological model[43] - interaction between individuals and social environment | Written narrative | Thematic analysis of written essays. |
| Leslie & Koblinsky (2017) [3] | USA | Women (n = 29)<br>Branch: Army (n = 19), Navy (n = 5), Air Force (n = 3), Marine Corps (n = 2)<br>Rank: Enlisted (n = 20), Officer (n = 8)<br>Deployed (n = 22) | Organismic Valuing Theory of Growth through Adversity[44] | Focus groups with semi structured interviews | Thematic analysis of focus groups |
| Libin et al. (2017) [58] | USA | Women (n = 2), men (n = 8)<br>*Characteristics for women only*<br>Age: 28 and 32<br>Branch: Army (n = 1), Air Force (n = 1) | Schlossberg's Transition Theory [45]; Role exit [46] | Semi-structured, in-depth interviews | Phenomenographic approach |
| Orazem et al. (2017) [2] | USA | Women (n = 42), Men (n = 58)<br>*Characteristics of full sample (men and women)*<br>Age: range 22–66 years, avg. 36<br>Branch: Army (n = 60), Air Force (n = 16), Navy (n = 13), and Marine Corps (n = 11)<br>Rank: Enlisted (n = 82), officers/warrant officers (n = 18) | Theories of identity formation [47] Erikson's stages of psychosocial development [48] | Written narrative describing reintegration difficulties | Thematic analysis |
| Ahern et al. (2015) [59] | USA | Women (n = 7), Men (n = 17)<br>*Characteristics of full sample (men and women)*<br>Age range: 22–55.<br>Branch: Air Force (n = 2), Army (n = 8), Army National Guard/Reserves (n = 5), Marines (n = 5), Navy (n = 4)<br>Time since discharge: 75% within the last 4 years | Homecoming Theory[49] | Semi-structured, in-depth interviews | Thematic analysis |

(*Continued*)

**Table 1.** (Continued)

| Author Publication Year | Country | Participant Characteristics | Primary Underlying Theories | Method of Inquiry | Type of Qualitative Analysis |
|---|---|---|---|---|---|
| Burkhart & Hogan (2015) [62] | USA | Women (n = 20) Age range 23–65 years, average 45 years Branch: Navy (n = 7), Air Force (n = 6), Army (n = 5), Marines (n = 2). Rank: enlisted (n = 10), Commissioned officers (n = 10) Length of Service 2–30 years, average 16.7 years. | Erikson's stages of psychosocial development[48] | Semi-structured, in-depth interviews | Grounded Theory |
| Mankowski et al. (2015) [60] | USA | Women (n = 18) Age range: 27–63, median age 43.5 Rank: E 3–5 (n = 6), E6-9 (n = 8), O 3–5 (n = 4) All had been deployed 2+ times. | Bridge theory: military as a bridge to adult roles [50,51] | Semi-structured, in-depth interviews | Grounded theory, constant comparative method |
| Koenig et al. (2014) [61] | USA | Women (n = 14), Men (n = 17) *Characteristics of full sample (men and women)* Age: Median 30 Rank: Enlisted 20 (64.5%), Officer 11 (35.5) All had been deployed to Iraq or Afghanistan. | Faulkner: Stages in the re-entry from military to civilian life[52]; | In person semi structured interviews | Grounded practical theory with interpretive analysis/ theme-oriented discourse analysis. |
| Demers (2013) [7] | USA | Women (n = 17) Age range 22–43, median age 29 Branch: Army (n = 10), Marines (n = 3), Navy (n = 4). Rank: Enlisted (94%) | Van Gennep's Rite of passage [53] | Focus groups with semi-structured interviews | Critical interpretive approach informed by hermeneutic phenomenology |

n = 80 themes and n = 91 subthemes exploring the transition process between the military and civilian environment and the role of gender during military service and post-service. Language regarding gender and its role in transition is largely taken from the studies themselves noting that gender incorporates cultural, social, and institutional meaning prescribed to one's sex. A discussion of feminist theory is beyond the scope of this review; however, we acknowledge the vast and evolving theories of gender, particularly how gender is operationalized within a military institution [35,72]. The interrelation of themes and subthemes is provided in Table 2. Explicit detail of the themes and subthemes extracted are available in the supplemental material (S2 File).

## Synthesis of findings

Our analysis revealed a strong link between in-service experiences and the transition to civilian life, especially for female veterans. Although the review focused on post-military outcomes, female service members' transition narratives consistently referenced their time in the Armed Forces. These accounts highlighted how gender-specific challenges spanned both military and post-service life.

### In-service military experience

**Indoctrination: Military identity and norms.** Following enlistment in the military, service members underwent an intensive process of indoctrination [68,69,73]. Participants described an initial "culture shock" as they adjusted to the new military environment[61,62].

**Table 2. Qualitative themes identified in the systematic review.**

| Articles | In Service Military Experience | | | | | | Post-Military Transition | | | | |
|---|---|---|---|---|---|---|---|---|---|---|---|
| | Military Identity | Military Norms | Feminine at odds with Masculine Ideal | Gender inequality: Lack of equipment & services | Gender Discrimination & Misogyny | Military sexual harassment and sexual assault | Loss of a military identity | Military Civilian Divide | Loss of Purpose | Female Veteran Identity | Civilian feminine norms at odds with military norms |
| Raabe et al. (2024) [63] | X | X | | | | | X | X | X | | |
| Barrington et al. (2023) [67] | X | X | X | | X | | X | X | X | | |
| Murray & Cancio (2023) [64] | X | X | | | | | X | X | X | | |
| Rattray et al. (2023) [71] | X | X | X | X | X | X | X | X | X | X | X |
| Barnett et al. (2022) [69] | X | X | | | | | X | X | X | | X |
| Eichler (2022) [68] | X | X | X | X | X | X | | | | X | X |
| Guthrie-Gowerm & Wilson-Menzfeld (2022) [70] | X | X | | | | | X | X | | | X |
| Laferty et al. (2022) [70] | | X | | X | X | X | | | | X | X |
| Boros et al. (2021) [56] | X | X | X | | X | X | X | | | | |
| Daphna-Tekoah et al. (2021) [65] | X | X | X | | X | X | | | | | |
| Sayer et al. (2021) [57] | X | X | | | X | X | | X | X | X | |
| Leslie & Koblinsky (2017) [3] | X | X | X | | | X | | | | | X |
| Libin et al. (2017) [58] | X | X | | | | | X | | | | |
| Orazem et al. (2017) [2] | X | X | | | | | X | X | X | | |
| Ahern et al. (2015) [59] | X | X | | | | | X | X | X | | |
| Burkhart & Hogan (2015) [62] | X | X | X | | X | X | X | X | X | | X |
| Mankowski et al. (2015) [60] | X | X | | | | X | | | | | |
| Koenig et al. (2014) [61] | X | X | | | | | X | X | X | | |
| Demers (2013) [7] | X | X | X | | X | X | X | X | | | X |

Military norms were introduced through the emotionally and physically intensive training, where all individuals were expected to cast off effeminate qualities and exhibit the desired, stereotypically, masculine norms [56,74,75]. Studies described how both men and women must adjust to the hegemonic masculine norms of the military[68]. Prized qualities reported included aggressiveness, physical and emotional toughness, dominance, bravery, competence, service to fellow soldiers, loyalty, and technical expertise[73,75,76]. Womanhood, femininity, and non-heteronormativity were reported to be seen as weak and subordinate to military hegemonic masculine norms [55,76]. Service members shifted their values and beliefs to reflect those supported by the military institutions, which were in turn, reinforced and solidified through their tenure in the military and the execution of their military duties [73].

**Impact of gender.** *Feminine at odds with masculine ideal.* Servicewomen in the studies described their gender as being in direct conflict with the stereotypically masculine norms of the military. They were forced to reconcile their identity as a female, a soldier, and for some a partner and mother, which were potentially in tension with one another [74]. Women in the studies described a variety of strategies used to adapt to a highly masculine military environment, including taking on the more masculine identities of their male counterparts, reducing outward feminine mannerisms, changing the way they moved, adjusting the tonality or content of speech, and being permissive of, or even joining in with the dominant humour [7,56,62,67,77]. Despite these efforts, many women expressed feeling unable to fully integrate and noted the stress of constantly adjusting themselves to fit the masculine norm [7,65,68]. Eichler (2022) noted that as women attempted to assimilate by reducing their femininity, they were paradoxically "hyper-visible" as a minority member in the Armed Forces[68], highlighting the "otherness" of being female which remained a barrier to attaining full group membership [68].

*Gender inequality, misogyny and discrimination.* Gender inequality, discrimination, and hostile work environments where men are viewed and treated as superior, were described as pervasive within the Armed Forces [7,55,56,62,68]. Many women voiced the need to work harder than their male counterparts to be perceived as valued members of the military [7,55,56,65,68]. Women experienced continuous pressure for high-level performance in a bid to be seen as equal to men [7,68]. Mistakes were seen as confirmation of their inadequacy and reflected poorly on other female service members [7,68]. Some women reported that they were assigned menial tasks, and their career progression was constrained because of their gender [68]. Gender inequality was also apparent in the lack of physical preparedness in the military for female service members with military equipment and clothing provided to women deemed inadequate to meet their needs [55,68]. These items were often ill-fitting and unsuited to the female statue, often resulting in injury [55]. These biases further extended to gender-specific health needs with inadequate health services and medical expertise to manage genitourinary conditions and reproductive health concerns specific to women [12,55].

Female service members also experienced sexual objectification, sexual harassment and the threat of sexual assault [33,55,65,68]. These threats increased during periods of deployment, requiring a heightened sense of awareness and compounding trauma experienced by service members whilst on deployment [55,65]. While both discrimination, harassment and assault were reported as pervasive across the studies, women noted differences depending on a history of deployment, service branch, rank, employment, and ethnicity [62]. For example, female service members working in the healthcare sector and women of higher rank reported discrimination to a far lesser degree than women of lower rank, women in forward-facing or combat positions, and women of colour [55,56,62].

## Post-service experiences

**Loss of military identity.**   For many service members, whether male or female, transitioning out of the military resulted in a profound sense of identity loss and required a renegotiation of one's identity within civilian society [59,64,67,70]. Re-entry to civilian life was often accompanied by a "reverse culture shock" as ex-service personnel constructed a new post-military identity and adjusted to the norms and expectations of civilian life [61,70].

A portion of male and female veterans expressed a sense of bereavement at losing their military identity and struggled with constructing a new identity in its place [2,7,56,64,70]. Some veterans described living two separate lives—one military and one civilian [7,62,70]. At one end, veterans actively fostered their military identity and rejected any alignment with a civilian identity [58]. On the other, veterans leaned into the civilian identity, over time fading their military identity [58]. However, most male and female veterans in the included studies occupied a middle realm, balancing their military identity with a civilian identity while constructing a new veteran identity [7,56,58,62–64].

**Military–civilian divide.**   Both male and female veterans routinely described the large chasm between military and civilian norms [2,7,59,61,63,67]. Veterans described feeling unable to be themselves around civilians due to differences in dress, mannerisms, humour, language and being perceived as inappropriate or aggressive [2,57,70]. These differences in military and civilian social norms impeded upon building meaningful relationships with civilians [2,7,57,59,61,69,70]. Some veterans found the detachment from civilians so profound that they were unable to reconnect with previously meaningful networks, such as childhood friends [62,70]. The lack of social connection left veterans detached and isolated from both their military and civilian community, often leaving them to navigate the hurdles of transition alone [2,7,59,61,69,70].

In workplace settings, the tension between military and civilian norms was especially stark. Both male and female veterans felt their unique skills were underutilized and underappreciated in the civilian workforce [2,57–59,63,78]. The unstructured nature of work, lack of clear policies, operating procedures, or clean lines of responsibility frustrated some veterans [60,62]. Conversely, civilian employers often perceived veterans as rigid and inflexible in their working style. Veterans were also frustrated by their work colleagues and felt many lacked accountability, discipline, punctuality, and motivation [62]. Many veterans found civilian employment mundane, slow-paced, and the civilian workforce lacking motivation [3,59,61,63,65,78]. These sentiments were even evident in fields with equivalents in the civilian sector, such as healthcare [2,61].

**Loss of purpose.**   Many male and female veterans described a general lack of purpose or meaning in the civilian environment [2,57,59,61,63,64,67,69,78]. Veterans described the mismatch between the slower pace of civilian life set against the fast pace of military life [61]. Some veterans objected to the lack of urgency displayed by civilians, even family members, and were frustrated by the absence of resolve, structure and routine in their day-to-day lives [2,59]. Paradoxically, veterans also commented on feeling overwhelmed by the amount of choice, fluidity, and lack of central hierarchy in the civilian environment [2,59,62]. Many veterans found executing general activities of daily living, such as appointments or errands, to be overwhelming [59].

**Impact of gender.**   The way women left the military impacted their post-service identity formation. Some female veterans felt forced out due to a lack of upward mobility, being sidelined after family leave, or when demands as a parent conflicted with the needs of the military [68]. Other women left following sexual discrimination, harassment, or assault. Those who left

the service earlier than desired often expressed feelings of betrayal from the military [7,56,59,62,65,68].

*Civilian feminine norms at odds with military norms.* Gender norms were described as imposing an unnecessary burden on women with many female veterans experiencing a discordance between feminine civilian and military norms [7,62,68]. The adjustments women made to adapt to the male-dominated military environment were often less well-received in civilian roles [62]. The expression of stereotypical feminine norms in the civilian environment, communicated through clothing, physical movement, mannerisms, and tonality of speech, informed all levels of interactions [62,68,70]. These included social rules for relationships, family dynamics, and workplace environment [68,70]. Female veterans described being perceived as aggressive or domineering [62]. These tensions were further complicated for women who had deployed or engaged in combat roles, as these experiences were at odds with the feminine ideal in civilian society [62,68].

Feminine norms associated with divisions of labour continued to apply pressure on women serving in the military, with the role of caregiving disproportionately falling to women. Just over half of all partnered female service members were in a dual-serving household, with family demands often cited as a predominate catalyst for exiting the armed forces [68]. Given the elevated pressures and demands, partnerships often ended in separation [68].

*Female veteran identity and needs.* Female ex-service personnel were not able to as easily embrace a veteran identity compared to their male counterparts [68]. Women noted the inherent tension between their multiple identities as warrior, spouse and mother compared to the more concordant masculine identity of warrior, provider, and father [68]. These sentiments were also confirmed through their interactions with civilians and even healthcare providers. Female veterans noted a lack of recognition of their military service or a minimization of their role in the military by both civilians and military healthcare providers [33,55,68].

Female veterans noted that healthcare and support services specific to their needs were routinely unaddressed [33,55]. They were often dismissed when seeking care for misunderstood chronic conditions, including fibromyalgia, chronic fatigue syndrome, and genitourinary concerns [33,55]. The potential connection between these conditions and military service were often underrecognized or negated [55]. Women also reported several reproductive challenges that they related to their exposures during deployment [55]. Women noted that these emerging and chronic conditions, such as urogynaecological, fatigue, migraines, and neurological disorders were often dismissed or misunderstood even among providers familiar with the veteran population (e.g. Veterans Affairs, US) [33,55].

**Consequence of military sexual trauma.** The negative consequences of Military Sexual Trauma (MST) were prominent themes that emerged across studies [7,33,55]. While no study directly asked about past experiences of harassment or assault, many female veterans voluntarily shared their experiences. The prolonged negative effect of MST extended beyond veterans mental and physical health but also affected their personal, social, and professional spheres [2,7,33,55,60,62,65,78]. Female veterans spoke about the adverse mental health effects, the avoidant behaviour, irritability, depression, anxiety, and PTSD resulting from MST [7,33,55]. Women also spoke of the lack of accountability, retaliation from reporting, and the sentiment that the military placed unit cohesion above the safety of female soldiers, often pushing women out of military service [60,62].

Of note, no male veteran participants provided accounts of sexual violence. Sexual violence and assault occur at a higher prevalence among female service members compared to male service members, however, male service members are less likely to report sexual violence [79].

## Discussion

The transition from military service to civilian life was particularly complex for women as they navigated the intersection of their military identity, gender-specific needs, and civilian expectations. Many female veterans encountered barriers in accessing appropriate healthcare, finding suitable employment, balancing family needs, and adjusting to civilian social norms. The adaptations made during service, including the adoption of more masculine traits often required in military environments, had to be recalibrated for the civilian world, which operated under different norms and expectations. This additional layer of adjustment was particularly challenging, as traits that were advantageous in the military did not always translate effectively to civilian professional or social contexts.

Our analysis also revealed a profound interconnection between in-service military experiences and the transition out of military service. For women, gender-specific challenges permeated both their military and post-military lives. Moreover, the transition process was often compounded by the lingering effects of trauma and discrimination.

The Five Eyes (FVEY) nations have actively pursued strategies to dismantle exclusionary policies and combat discrimination against female service members. These strategies include lifting bans on pregnancy, allowing women to serve in combat roles, and promoting equality for previously marginalized groups, including LGBTQ+ individuals. The FVEY nations have also advanced proactive policies, implementing more flexible working arrangements, enhanced parental leave, targeted mentorship programs for career development among female service members, and improved facilities and equipment for women [10,24,80–82]. These policies demonstrate the FVEY commitment to attracting and retaining female service members.

While these policy changes represent significant progress, their implementation has not been without challenges. Reports of gender-based discrimination, harassment, and assault in the AF persists [24,83,84]. This bears out in our findings, which revealed persistent gender discrimination, harassment and assault within the military and varying degrees of institutional acceptance of this behaviour.

The persistence of these issues has far-reaching consequences on the health and well-being of female veterans. While the exact prevalence of sexual harassment and assault is not known, it is clear that sexual harassment is widespread and sexual assault not uncommon [85]. Addressing gender-based discrimination, harassment, and assault in the military is not only an ethical necessity, but also a critical factor in improving overall health outcomes for service members and veterans. While female veterans generally report higher rates of mental and physical health comorbidities, research has demonstrated that these health disparities are strongly correlated with negative gender-based experiences during service [86]. Eliminating these harmful behaviours would potentially reduce the prevalence of mental and physical health issues among female veterans and improve their transition outcomes.

Given the challenges experienced by female service members, it is unsurprising that women often have shorter military tenures than their male counterparts and are more likely to leave before completing their terms of service [24,87]. This trend is particularly concerning as early service leavers (ESLs) tend to experience worse physical and mental health outcomes, lower financial stability, and housing insecurity [88]. Many women in the studies reported feeling compelled to exit the military out of necessity rather than choice. Factors contributing to early departure encompassed a range of challenges, from physical injuries and family responsibilities to gender-based discrimination, harassment, and sexual assault.

Changes in military identity emerged as a salient theme for all veterans transitioning out of the Armed Forces, with individuals struggling to varying degrees with this shift. Our analysis revealed that the formation of military identity among female veterans was often attenuated,

influenced by experiences of gender-based discrimination, military sexual trauma (MST), and the complexities of balancing roles as partners, mothers, and soldiers. Broader research has shown that veterans with a strong military identity often face more difficulties in transition, particularly in forming community connections and social networks [88]. Factors linked to stronger self-reported military identity include being male, longer service duration, and expectations of an extended military career [88,89]. Military identity exerts a strong effect on the transition process [88,89]. However, the specific impact of gender on identity transformation remains less clear, highlighting the complex interplay between gender, military experiences, and identity formation during transition.

Female veterans in the reviewed studies also noted the difficulty in accessing health services which were specific and sensitive to the needs of female veterans. They pointed to various barriers, including a lack of knowledge of what services were available, how to access these services and reluctance to access male-dominated support services [90]. Many of the female veterans also noted that their military service was unacknowledged or minimized, and there was poor recognition of service-related physical injuries. Our findings are consistent with previous literature showing continued challenges in accessing to sex and gender appropriate mental and physical healthcare, particularly in male-dominated veteran health care setting [91,92].

It's important to note that the challenges women face in the military are not unique and share notable parallels with those encountered in other male-dominated professions, particularly law enforcement and male-dominated emergency responders [93–95]. Across these sectors, women confront a spectrum of obstacles, including the need to adapt to pervasive masculine norms, endure discriminatory humour, and navigate workplace discrimination. They also face persistent scepticism about their physical capabilities and experience constant pressure to prove themselves. These experiences significantly impact mental and physical well-being, with many women reporting increased stress and burnout [93,94]. Despite policy changes in both the military and law enforcement aimed at addressing these issues, progress has been incremental, and retention of women in these professions remains a significant challenge [24,96].

Our findings suggest that effectively supporting female veterans requires recognizing their diverse military experiences, as these significantly influence their post-service needs. Support should include specialized career counselling to translate military skills to civilian environments, holistic programs addressing family needs, and targeted mental and physical health services sensitive to women's specific requirements. Moreover, support services should acknowledge and address the unique identity shifts female veterans undergo, assisting them in navigating the complex balance between their military-forged identities and civilian gender expectations.

## Major gaps

There is an emerging body of evidence examining the military-to-civilian transition among women [17]. However, numerous gaps in understanding remain. From the reviewed studies, there are three areas where knowledge is notably lacking: the role of a military and veteran identity among women, the long-term detrimental effects of discrimination and harassment, and the evaluation of health needs and services specific to female service members and female veterans.

The role of military identity in female veterans' transition processes is an understudied area. While existing literature has examined military identity and its impact on transition among male veterans, revealing both facilitating and inhibitory effects there is a notable paucity of research focusing on female veterans' experiences [89]. This gap is particularly

significant given the unique challenges women face, including the navigation of multiple roles, higher rates of discrimination, and the frequent minimization of their military experiences post-service.

Second, given the pervasiveness of gender discrimination and harassment, the long-term social, mental, physical, and economic costs to service members should be examined. It is critical to understand how discrimination based on any characteristic, such as gender, ethnicity, or religious affiliation, may undermine force readiness and unit cohesion.

Third, a more comprehensive assessment of the diverse social and health needs and enhanced services is required. Female service members and veterans spoke to a wide range of physical and mental health needs that vary according to individual needs and preferences, life stage, and environment. A broader conceptualization of veteran women's health will be required to address these needs effectively. This understanding should encompass concerns such as reproductive health, perimenopause and menopause, cardiovascular disease, cancer, autoimmune disorders, and mental health [97,98].

## Strengths and limitations

This review had a number of strengths. A comprehensive search strategy was developed in collaboration with a data librarian. We used an expanded set of controlled vocabulary terms and keywords were used across several databases to ensure relevant studies were identified. The review builds upon previous quantitative work where the synthesis of qualitative data allows for a richer understanding of the female veteran experience by finding the commonalities and differences in the veteran experience.

There are some limitations with the review. First, approximately three-fourths of the studies were from the US. While in-service experiences and issues related to identity are likely to be similar, social services and healthcare provisions in the civilian sector may differ across the represented countries. Second, there is a diversity of approaches to qualitative synthesis each with inherent limitations and biases. We included all studies, regardless of differing underlying methodologies (phenomenology, grounded theory, narrative analysis, etc.). Third, while there was considerable similarity in themes across the studies, these narratives may not encompass all the experiences of female veterans, nor do they capture the experience of various population by age or ethnicity.

## Conclusion

This review found that female veterans confront many stressors similar to their male counterparts when leaving the Armed Forces, while also experiencing some notable differences. The majority of studies reported that female service members encountered varying degrees of gender discrimination, misogyny, and gender inequality both during and post-service, with MST continuing to be a significant concern during active duty. As women enter the Armed Forces in greater numbers over the coming decades, research must advance our understanding of the barriers to successful transition for female service personnel, particularly beyond the US context. For meaningful progress, policymakers and the Armed Forces must be willing to confront and address the cultural legacy and entrenchment of gender inequality, discrimination, and MST. This approach is crucial for improving the experiences of women both during their service and in their transition to civilian life.

## Supporting information

**S1 File. Joanna Briggs Institute (JBI) qualitative critical appraisal checklist for qualitative studies.**
(DOCX)

**S2 File. Themes and subthemes.**
(DOCX)

**S3 File. Search strategy.**
(DOCX)

**S4 File. Selected quotes.**
(DOCX)

**S5 File. Data.**
(XLSX)

## Author Contributions

**Conceptualization:** Laura Rafferty, Bethany Croak, Neil Greenberg, Anne Stagg, Nicola Fear, Sharon Stevelink.

**Data curation:** Alexandria Smith.

**Formal analysis:** Alexandria Smith, Laura Rafferty, Rafiyah Khan.

**Funding acquisition:** Laura Rafferty, Bethany Croak, Neil Greenberg, Nicola Fear.

**Investigation:** Alexandria Smith, Laura Rafferty, Bethany Croak.

**Methodology:** Alexandria Smith, Laura Rafferty.

**Project administration:** Alexandria Smith.

**Software:** Alexandria Smith.

**Supervision:** Laura Rafferty, Marie-Louise Sharp.

**Validation:** Alexandria Smith, Laura Rafferty, Bethany Croak, Rafiyah Khan.

**Visualization:** Alexandria Smith, Laura Rafferty.

**Writing – original draft:** Alexandria Smith, Laura Rafferty, Bethany Croak.

**Writing – review & editing:** Alexandria Smith, Laura Rafferty, Bethany Croak, Neil Greenberg, Rafiyah Khan, Victoria Langston, Marie-Louise Sharp, Nicola Fear, Sharon Stevelink.

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
