## [Decision Letter · Decision Letter 0]

3 Jun 2024

PONE-D-24-15468A Systematic Review of Military to Civilian Transition: The Role of GenderPLOS ONE

Dear Dr. Smith,

Thank you for submitting your manuscript to PLOS ONE. After careful consideration, we feel that it has merit but does not fully meet PLOS ONE’s publication criteria as it currently stands. Therefore, we invite you to submit a revised version of the manuscript that addresses the points raised during the review process.

We look forward to receiving your revised manuscript.

Kind regards,

Darrell Eugene Singer, M.D., M.P.H.

Academic Editor

PLOS ONE

Journal Requirements:

"This project was supported by a grant from the Forces in Mind Trust (FiMT) 2202"

"No competing interests"

4. We noted in your submission details that a portion of your manuscript may have been presented or published elsewhere. [Available for preprint on medRxiv (MS ID#: MEDRXIV/2024/303195)] Please clarify whether this [conference proceeding or publication] was peer-reviewed and formally published. If this work was previously peer-reviewed and published, in the cover letter please provide the reason that this work does not constitute dual publication and should be included in the current manuscript.

5. In the online submission form, you indicated that [All underlying data is available upon request from researchers. The search strategy, including the exact search terminology will be provided in supplementary data.]. 

7. Please include your tables as part of your main manuscript and remove the individual files. Please note that supplementary tables (should remain/ be uploaded) as separate "supporting information" files

Additional Editor Comments:

I concur with the reviewers that this is an important study and compliment you and your co-authors on your efforts. It is a privilege to edit your submission.

I have attempted to summarize the reviewers’ comments and provide specific guidance for a revision, however, there are several questions on the focus and methodology that may generate further discussion. Please also review and address the comments made by each reviewer that were not addressed in the summary. As this manuscript requires a major revision, the subsequent version may require additional reviews and/or edits.

1. Grammar, syntax, and formatting:

1a. Please insert page and line numbers (continuous) into the manuscript. https://journals.plos.org/plosone/s/submission-guidelines#loc-page-and-line-numbers

1b. The manuscript contains grammatical, syntax, and formatting errors that must be corrected. PLOS ONE does not copy-edit accepted manuscripts and directs Academic Editors not to provide copy-edits, however, Academic Editors and reviewers may point out some examples to assist the authors. https://journals.plos.org/plosone/s/submission-guidelines. Examples include the use (or not) of hyphens: “military to civilian transition” versus “military-to-civilian transition” and “post transition” versus “post-transition.” The overall article requires light editing on the use/absence of articles and prepositions.

1c. Consistency in terminology/references is helpful: examples include “gender discrimination” vs “gender-based discrimination.” The first two sentences of the second paragraph of the background section are somewhat clunky- the first sentence mentions increases in the “number of women” and the second sentence mentions prevalence. The two could be merged and streamlined. A similar sentence is contained in the conclusions section states “As women continue to enter the AF in greater numbers over the coming decades, research must advance our understanding of the barriers to a successful transition for female service personnel, particularly beyond the US context.” Suggest the consistent use of prevalence or proportion.

1d. Please review reference formatting and font use.

2. Introduction:

2a. Please provide an expanded description of why the military-civilian transition is an important aspect of military and veteran population studies. This could include the magnitude of the problems or build on the authors’ statements regarding the increasing prevalence of women in the military. A brief discussion of the definitions and use of the terms “female” and “woman” may be applicable.

2b. The introduction section should include an overview of similar studies (systematic reviews or meta-analyses) of qualitative studies and discuss any findings and/or gaps not answered by previous studies. What will your study add to the discussion? The last sentence in the second to last paragraph of the introduction provides a hint; possibly this should be combined with the last sentence of the section. “However, there remains ample room for improvement in policies and programs to support female veterans and in order to do this effectively we need further understand how women’s military experiences and post transition experiences may differ compared to those of men.”

2c. Please mention why the review is limited to the Five Eyes nations.

3. Methodology:

3a. Please include the study’s registration # in the manuscript.

3b. Seven databases were listed in the manuscript and Appendix C. Were other databases queried? From the methodology section: “A multi-database search was conducted, including Medline, Embase, PsycINFO, Pubmed, Global Health, Web of Science, and EBSCO.” If not, I recommend editing for specificity, e.g. “A multi-database search was conducted of Medline, Embase, PsychINFO….” If other databases or literature sources were accessed, please expand the methodology to include how these were addressed.

3c. Please provide a deeper description of the inclusion/exclusion criteria. Specifically, the availability of full text and the first screener’s criteria for inclusion/exclusion into the study. Details could include what topics (medical, psychological, etc.) and when those experiences occurred (During service, post service, either, both)?

3d. Please provide an expanded description/definition of the framework analysis elements: transcription of verbal data to text; familiarisation; coding; developing an analytical framework; applying the analytical framework; charting data into the matrix; interpreting the data so that the study could be replicated. Why was a seventh element not included?

3e. Systematic reviews are recommended to have multiple reviewers, but the recommendations regarding the timing and use of the additional reviewers are inconsistent. Please describe how bias was avoided in your study, particularly the single screener and how possible discordancy between the two reviewers was handled.

4. Results:

4a. How do “results” vs “study findings” differ to require separate headers? Please provide a rationale for why. The difference is apparent- one appears to present descriptive findings; the second appears to present the qualitative analysis; however, both are “results.” Suggest a header of “Results” with sub-headers of “descriptive results vs. synthesized” or the like.

4b. The experiences occurring during a female/woman’s service experience are important and certainly contribute to their post-service health condition. Please provide an expanded description of how the “In-Service Military Experience” section contributes to the study’s transition objectives.

4c. Please provide an expanded discussion of the qualitative appraisal and thematic information. Did the investigators assess inter-nation differences?

5. Discussion:

5a. The discussion seems to repeat the results section. Please provide an expanded discussion on whether the findings of your study confirm, support, or counter previous studies and/or findings. Are there differences in transitioning populations (gender, age, race, other)?

Again, thank you for your work and patience with the editorial process. I look forward to your response.

Reviewers' comments:

Reviewer's Responses to Questions

**Comments to the Author**

1. Is the manuscript technically sound, and do the data support the conclusions?

Reviewer #1: Partly

Reviewer #2: Yes

2. Has the statistical analysis been performed appropriately and rigorously? 

Reviewer #1: N/A

Reviewer #2: N/A

3. Have the authors made all data underlying the findings in their manuscript fully available?

Reviewer #1: Yes

Reviewer #2: Yes

4. Is the manuscript presented in an intelligible fashion and written in standard English?

Reviewer #1: Yes

Reviewer #2: Yes

5. Review Comments to the Author

Reviewer #1: This systematic review aimed to identify and synthesize qualitative findings exploring the impact of gender during the military-civilian transition. The military-civilian transition is a time period of vulnerability for military service members, thus is a noteworthy area of inquiry across multiple countries. The abstract is clear and concise and results reported are within the scope of the aim of the review. However, there are noteworthy limitations in other sections of the manuscript.

Introduction

1. Need additional information on why the military-civilian transition is a transition that should be a focus when studying military and veteran populations.

2. No specific policies mentioned across countries considered in the review.

3. Does not provide and overview other scoring reviews to identify additional gaps in other publications using this methodology. Mentions only one review.

Methods

1. Include registration #.

2. Was grey literature considered in the review? If so, how was it addressed?

3. Explain why availability in full text was an inclusion criteria considering interlibrary loan and other strategies are available to access publications. Consider availability at institutions of the large group of authors of this manuscript.

4. Explain Framework Analysis and rationale for use in this review.

Results

1. How do "results" and "study findings" differ? One heading with limited subheadings seems appropriate to guide the reader.

2. The In-Service Military Experience section is beyond the aim of the review. It is not military-civilian transition focused. Seems more appropriate for the introduction to provide a rationale for the focus of the review. The space used by this section can be better used to include more elements of the manuscript related to framework analysis.

3. Qualitative appraisal discussed minimally as well as thematic information in supporting documentation.

4. Are there differences by country?

Discussion

1. Limit number of subheadings

2. Seems repetitive of the results and themes

3. Does not link findings of this review to the larger literature (i.e., both qual and qual) related to military-civilian transition focused on gender and/or focused on the military and veteran population in general.

4. Does the review findings confirm, support, and/or extent prior findings? Suicide theories? Military-civilian transition frameworks given the themes related to sense of belonging, loss of purpose, loss of military identity?

5. Link to to military research findings on gender discrimination instead of "workplace harassment" in general.

6. Is the major gaps section supposed to be one paragraph?

7. Qualitative methods such as grounded theory have specific outcomes related to theorizing. Do included studies with specific qualitative methodology contribute to theory or other outcomes related to women military service members and veterans?

8. Are there international implications? Adoption of policy and procedures for military women and veterans internationally?

Overall

1. Focus results and discussion within the scope of the review. Identify gender differences and differences among women (e.g., age and racial differences, military service branch, active vs reserve).

2. Edit more carefully to address missing articles (e.g., a, and the) and prepositions.

3. Review reference formatting to be consistent with authors' instructions.

4. Correct use of two different fonts and use terminology consistently throughout the manuscript (e.g., gender discrimination vs gender-based discrimination.

Reviewer #2: Outstanding work, and thank you for allowing me to review this study.

This paper is a scoping review regarding the experiences of female veterans through the military to civilian transition. The review employs a framework analysis to collect general themes and subthemes from the accepted literature, and then synthesizes the findings within those overarching theme groups. Examples of these themes include loss of purpose and discordance of gender norms. Author(s) conclude that the women face significant stresses and disparities compared to male counterparts, especially post-service.

The overall methodology appears appropriate for the research question and what the authors are looking for. The “Inclusion/Exclusion Criteria” properly fits, considering the general nature of experiences of military to civilian transition. The process of study selection and screening is adequately noted. Data extraction, synthesis, and results are appropriate and work well within the overall scoping review. The supplemental charts and the JBI critical appraisal are properly completed and there are no issues regarding those features. The articles are well summarized and themes are well presented.

In essence, the article is almost acceptable. However, I do advise a couple of revisions or explanations, not only for a complete research article, but also for duplication purposes.

Major Revisions/Comments

1. I do advise more details for inclusion/exclusion factors, especially for purposes of study duplication. Although the details of participants, study type, and theme of search are presented, there should be specific mentioning for what qualifies during the screening process. For instance do these experiences need to be regarding a certain topic (medical, psychological, etc.) and when can those experiences have occurred (During service, post service, either, both)?

2. There is a notable inquiry as to why only one reviewer conducted initial screenings of titles and abstracts - Many scoping review studies often utilize multiple reviewers throughout the entire screening process. In any case, it may be worth mentioning potential bias/lack of consensus during initial screening.

Minor Revisions/Comments

1. First sentence of “Inclusion/Exclusion Criteria” - May want to break up the sentence a bit. This is mostly to reduce ambiguity of inclusion factors.

2. “Results” section - Please add the study methods within the included studies (interview, focus group, etc.) to detail more about what current literature is available

3. I do think there is a lack of consistency of themes/subthemes are worded within appendix B, which may provide difficulty in navigation.

6. PLOS authors have the option to publish the peer review history of their article (what does this mean?). If published, this will include your full peer review and any attached files.

Reviewer #1: No

Reviewer #2: No

---

## [Author Response · Author response to Decision Letter 0]

30 Sep 2024

I am pleased to submit the revised version of our manuscript titled “A systematic review of military to civilian transition: The role of gender” [PONE-D-24-15468] for consideration. We have carefully considered and addressed each of the reviewers' comments. The majority of the recommended changes have been implemented, which has significantly enhanced the quality and clarity of our manuscript. For suggestions that we were unable to incorporate, we have provided detailed justifications for our decisions.

We are confident that these revisions have substantially strengthened our manuscript. We sincerely appreciate your time and consideration and look forward to your response.

Detailed Responses to Reviewers' Comments PONE-D-24-15468

Grammar, syntax, and formatting:

1. Insert page and line numbers (continuous) into the manuscript. 

2. Correct grammatical, syntax, and formatting errors. 

3. Ensure consistency in terminology/references (e.g., "gender discrimination" vs "gender-based discrimination"). 

4. Review reference formatting and font use.

Response: 

1. Page and line numbers have been inserted continuously throughout the manuscript.

2. All grammatical, syntax, and formatting errors have been corrected. 

3. Terminology and references have been standardized throughout the document (e.g., consistent use of "gender-based discrimination"). 

4. Reference formatting has been reviewed and corrected, and font usage is now consistent throughout the manuscript.

Introduction: 

1. Expand on the importance of military-civilian transition in military and veteran population studies. 

2. Include an overview of similar studies and discuss findings/gaps not answered by previous studies. 

3. Explain why the review is limited to the Five Eyes nations.

Response: 

1. We have elaborated on the importance of military-civilian transition in military and veteran population studies. We've highlighted that the Five Eyes nations recognize their obligation to support veterans and families, ensuring they are not disadvantaged by their service. We've noted the UK's formalization of this commitment through the Armed Forces Covenant. We've emphasized that as women's representation in the military grows, understanding their unique transition challenges is critical for both supporting veterans and enhancing retention and recruitment efforts. (Line 120-126)

2. We've included an overview of similar studies, particularly previous scoping reviews focusing on quantitative studies related to physical and mental health outcomes. We clarified that our study differs by synthesizing qualitative literature examining the experience of military-to-civilian transition and the impact of gender, offering deeper insights into veterans' lived experiences. (Line 132-142)

3. We've explained that the review is limited to the Five Eyes nations due to their shared language and closely aligned intelligence and military policies, providing a cohesive context for examining veteran transition. (Line 116-118)

Methodology: 

1. Include the study's registration number in the manuscript. 

2. Clarify the databases used in the search. 

3. Provide a deeper description of inclusion/exclusion criteria. 

4. Expand on the framework analysis elements and explain why a seventh element was not included.

5. Describe how bias was avoided, particularly regarding the single screener and handling of discordancy between reviewers.

Response:

1. The study's registration number has been included (Open Science Framework registration: osf.io/5stuj). (Line 147)

2. We have clarified the databases used in the search. A multi-database search was conducted using Medline, Embase, PsycINFO, Pubmed, Global Health, Web of Science, and EBSCO. (Line 149-151)

3. We provided a deeper description of inclusion/exclusion criteria, detailing participant requirements, study types, and reasons for focusing on FVEY nations. (Line 160-172)

4. We explained that the transcription of verbal data into text was bypassed as we were extracting narratives from peer-reviewed studies. (Line 208-209)

5. We described our approach to avoid bias: initial screening by one reviewer using a conservative approach, followed by full-text reviews conducted independently by two reviewers with consensus required for study inclusion. (Line 176-182)

Results: 

1. Clarify the difference between "results" and "study findings" sections or restructure headers.

2. Expand on how the "In-Service Military Experience" section contributes to the study's transition objectives.

3. Provide an expanded discussion of qualitative appraisal and thematic information, including inter-nation differences.

Response:

1. We altered the headings of the results to include “descriptive results” in which we discuss the characteristics of included studies, such as date of publication, country, number of participants, analytic framework etc. “Study findings” was changed to “Synthesis of findings” to indicate reporting on the themes from the included studies. (Line 222, Line 271)

2. We included additional language at the start of the "In-Service Military Experience" to provide our rationale for focusing on in-service experiences. (Line 279-291)

3. We expanded the discussion of our approach to qualitative appraisal in the methods section and included a brief discussion of the international findings. (Line 224, Line 576 – 579, Line 964)

Discussion: 

1. The discussion seems to repeat the results section. 

2. Please provide an expanded discussion on whether the findings of your study confirm, support, or counter previous studies and/or findings. 

3. Are there differences in transitioning populations (gender, age, race, other)?

Response:

1. We addressed the repeating nature of the results section. (Full alteration of discussion section, Line 446-583)

2. We integrated previous transition studies into each of the paragraphs to confirm or counter previous findings (interspersed within discussion section Line 446-583)

3. We further discussed the inability to look at race, age and other demographics in the discussion section due to lack of demographic information on participants. (Line 582-585)

---

## [Decision Letter · Decision Letter 1]

11 Dec 2024

A Systematic Review of Military to Civilian Transition: The Role of Gender

PONE-D-24-15468R1

Dear Dr. Smith,

We’re pleased to inform you that your manuscript has been judged scientifically suitable for publication and will be formally accepted for publication once it meets all outstanding technical requirements.

Kind regards,

Darrell Eugene Singer, M.D., M.P.H.

Academic Editor

PLOS ONE

Additional Editor Comments (optional):

Thanks to you and your fellow authors for your collective patience and persistence- congratulations and well done.

Reviewers' comments:

Reviewer's Responses to Questions

**Comments to the Author**

1. If the authors have adequately addressed your comments raised in a previous round of review and you feel that this manuscript is now acceptable for publication, you may indicate that here to bypass the “Comments to the Author” section, enter your conflict of interest statement in the “Confidential to Editor” section, and submit your "Accept" recommendation.

Reviewer #2: (No Response)

Reviewer #3: (No Response)

2. Is the manuscript technically sound, and do the data support the conclusions?

Reviewer #2: Yes

Reviewer #3: Yes

3. Has the statistical analysis been performed appropriately and rigorously? 

Reviewer #2: Yes

Reviewer #3: N/A

4. Have the authors made all data underlying the findings in their manuscript fully available?

Reviewer #2: Yes

Reviewer #3: Yes

5. Is the manuscript presented in an intelligible fashion and written in standard English?

Reviewer #2: Yes

Reviewer #3: Yes

6. Review Comments to the Author

Reviewer #2: I thank the authors for their resubmission of their systematic review.

The authors adequately addressed my previous comments - The paper's methodology appears technically sound, and all inclusion/exclusion factors and potential for bias biases are properly explained.

The paper's new discussion section appears to properly describe the results in relation to current literature, properly expanding upon the military-to-civilian transition.

Please review paper for grammar and formatting, such as:

Missing periods line 115, line 469 line 567

Reference formatting Line 440

No other significant/major changes are noted.

Reviewer #3: I enjoyed reading the article. It is a well written article of original research on a topic of interest within the realm of military/veteran well-being among Five Eyes nations. The article is technically sound. The analysis, discussion and conclusion are appropriate given the findings presented and the acknowledged limitations. The authors have sufficiently addressed the feedback from the previous reviewers.

7. PLOS authors have the option to publish the peer review history of their article (what does this mean?). If published, this will include your full peer review and any attached files.

Reviewer #2: No

Reviewer #3: No

---

## [Editor Report · Acceptance letter]

8 Jan 2025

PONE-D-24-15468R1 

PLOS ONE

Dear Dr. Smith, 

I'm pleased to inform you that your manuscript has been deemed suitable for publication in PLOS ONE. Congratulations! Your manuscript is now being handed over to our production team.

Kind regards, 

on behalf of

Dr. Darrell Eugene Singer 

Academic Editor

PLOS ONE